# Microfluidic-Assisted Fabrication of Monodisperse Core–Shell Microcapsules for Pressure-Sensitive Adhesive with Enhanced Performance

**DOI:** 10.3390/nano10020274

**Published:** 2020-02-06

**Authors:** Xiangshen You, Bingsheng Wang, Shuting Xie, Lanhui Li, Han Lu, Mingliang Jin, Xin Wang, Guofu Zhou, Lingling Shui

**Affiliations:** 1Guangdong Provincial Key Laboratory of Optical Information Materials and Technology, South China Academy of Advanced Optoelectronics, South China Normal University, Guangzhou 510006, Guangdong, China; xiangshenyou@m.scnu.edu.cn (X.Y.); bestbs@m.scnu.edu.cn (B.W.); stxie@m.scnu.edu.cn (S.X.); lanhui.li@m.scnu.edu.cn (L.L.); hanlu@m.scnu.edu.cn (H.L.); wangxin@scnu.edu.cn (X.W.); guofu.zhou@m.scnu.edu.cn (G.Z.); 2International Academy of Optoelectronics at Zhaoqing, South China Normal University, Guangzhou 510631, Guangdong, China; 3School of Information and Optoelectronic Science and Engineering, South China Normal University, Guangzhou 510006, Guangdong, China

**Keywords:** microfluidics, droplet, microcapsule, pressure-sensitive adhesive, bonding

## Abstract

Microcapsule-based adhesives hold special properties offered by encapsulation via an interfacial shell. Microencapsulation provides the possibility of combining the materials with opposite properties for which co-existence is commonly difficult. In this work, we report on a high performance pressure-sensitive adhesive (PSA) based on monodisperse and size-controllable core–shell microcapsules, which are prepared from double-emulsion droplets constructed using microfluidic devices. Monodisperse microcapsules containing oxalic acid are prepared with a coefficient of variation (CV) size of <5% and the core-material encapsulation efficiency of >90%. The microcapsules and urea-formaldehyde resin are mixed to obtain capsules-based PSA. The overall size uniformity achieved from droplet microfluidics and the rigid interfacial shells from photopolymerized materials ensure high rupture efficiency and sufficient curing reaction during the process. The microcapsules with proper shell thickness can well encapsulate the core material with an even distribution in the center, separating the curing agent from the matrix resin to form a latent adhesive, which is released at the right place and the right time. The bonding strength of >0.7 MPa has been achieved for plywood boards bonding using the prepared PSAs. The capsule-based PSA could encapsulate active components to achieve an extended lifetime for storage, controlled release to achieve on-demand operation, and pressurized mechanical rupture for ease of use. These would be expected to promote the use of PSAs in modern industries such as micro- and nano-optoelectronic devices by further tuning the size and materials of the microcapsules.

## 1. Introduction

The encapsulation of materials is normally applied for isolation or protection, being evolved from nature, ranging from nanoscale to macroscale, from a cell to an egg [1]. The global microcapsule market was valued to be $6.3 billion in 2018 and projected to rise to $11.8 billion by 2023 [2]. Microcapsules have been applied in multiple fields such as self-healing materials, drug delivery, fragrance release, and gas adsorption [3,4,5,6]. From the perspective of these applications, the controlled response of microcapsules includes chemical, biological, mechanical, light, thermal, magnetic, and electrical stimuli [7]. In industry, the adhesives market was projected to grow from $41.47 billion in 2016 to $53.37 billion by 2021 [8]. Pressure-sensitive adhesives (PSA) based on microcapsules show the advantages of low price and easy operation, which have been applied in labels, adhesive tapes, the industrial processing of optoelectronic devices, etc. [9,10,11,12]. In PSA, partial components or the total formulation of adhesives can be encapsulated as the core materials, which are released by pressurized shell rupture to achieve on-demand bonding. 

Urea-formaldehyde (UF) resins are widely used in wood adhesives due to their easy process and low price [13]. The curing efficiency of UF resins is dependent on pH value; thus, an acidic curing agent is usually applied [14]. As a result, to assure on-demand rapid curing, a large amount of H^+^ should be provided by curing agents at once. On the other hand, the reaction time should be controlled at a proper scale to ensure proper bonding. Therefore, a desirable curing agent should be stable for a long time and enable the rapid curing of UF resins when released. To satisfy these requirements, encapsulation via core–shell capsules is preferred, which can well protect the reactive acid for a long time and quickly release it by simple pressing to achieve bonding performance as required [9]. 

Microcapsules are commonly prepared by emulsion polymerization, layer-by-layer assembly, phase separation, sol-gel encapsulation, spray-drying, fluid-bed coating, etc. [7,15,16,17]. These methods are faced with either low encapsulation efficiency of components or an inhomogeneity in microcapsule size, resulting in a low release ratio and poor controllability during the actual working process [18]. In PSA, we can imagine that polydisperse microcapsules would cause incomplete rupture once pressurized to a constant distance, thus insufficiently releasing encapsulated components. Therefore, in practical applications, a controllable size and uniformity in both the diameter and shell of the microcapsules are required for high efficiency PSA. 

Droplet microfluidics has been widely applied for preparing droplets/capsules due to its precise control over size and uniformity [19,20,21,22]. Being differentiated from conventional methods, the microfluidic technique utilizes hydrodynamic instability to form double-emulsion droplets in a controlled manner [23,24]. Due to these advantages, there have been many studies on the microfluidic preparation of droplets/capsules for drug delivery, cell encapsulation, self-healing materials, and catalysts [25,26,27,28].

In this work, we constructed high-quality monodisperse core–shell microcapsules for PSA from double-emulsion droplets via the microfluidic technique. Monodisperse water-in-oil-in-water (W/O/W) double-emulsion droplets with controllable size were created by using a glass-capillary co-flow microfluidic device, with an oxalic acid (initiator) glycol solution as the inner phase, an ethoxylated trimethylolpropane triacrylate (EPTPA) prepolymer solution as the middle phase, and an aqueous solution containing 4 wt% PVA and 30 wt% glycerol as the outer phase. Upon UV irradiation, EPTPA were cross-linked to form the shell to enclose the core, forming a microcapsule. The size and shell thickness of the microcapsules could be precisely controlled by adjusting the fluidic flow rates. The stability of the obtained PSA microcapsules in water with different shell thicknesses was also explored. PSA was prepared by mixing the constructed microcapsules with UF resin powder and water. When two wood boards coated with PSA were brought closely via a perpendicular pressure, rigid shells of the microcapsules were ruptured to release oxalic acid to initiate a cross-linking reaction with UF, forming glue to achieve bonding performance. Stable encapsulation from microcapsules protected the core materials from leaking and ensured their lifetime. The microcapsule’s uniform size (diameter and shell thickness) ensured high proportion release once pressing and rapid curing. Moreover, the precoating and controlled release process could achieve convenient, quick, and efficient bonding.

## 2. Materials and Methods 

### 2.1. Materials

Deionized (DI) water (18.25 MΩ·cm at 25 °C) was prepared using a Milli-Q Plus water purification system (Sichuan Wortel Water Treatment Equipment Co., Ltd, Sichuan, China). Urea-formaldehyde resin powder was purchased from Senbang Chemical Co., Ltd. (Hebei, China). Rubber plywood with a water content of 8.0 wt% was obtained from Weiling Wood Co., Ltd. (Shanghai, China). Ethoxylated trimethylolpropane triacrylate (ETPTA), polyvinyl alcohol (PVA, M_w_ = 13,000), and trichlorosilane were purchased from Sigma-Aldrich (Shanghai, China). Oxalic acid was purchased from Zhiyuan Chemical Reagent Co., Ltd. (Tianjin, China). Methyl orange was purchased from Guangzhou Chemical Reagent Factory (Guangdong, China). The initiator of 2-dimethoxy-2-phenylacetophenone was bought from Heowns Biochem Technologies Co., Ltd. (Tianjin, China). 

### 2.2. Preparation of Microfluidic Device

The capillary microfluidic device consisted of coaxially assembled cylindrical and square glass capillaries (Sutter Instrument, Novato, CA, USA), which were connected using a tee connector. Cylindrical capillaries with an outer diameter (OD) of 1.0 mm and inner diameter (ID) of 500, 300, and 200 μm were used to accommodate different stages of emulsification for preparing droplets. The end of the injection capillary tube was tapered to 20–40 μm in tip diameter using a capillary puller (P-1000, Sutter Instrument, Novato, CA, USA). A larger tip diameter (60–80 μm) was obtained by using sandpaper. Cylindrical capillaries were rinsed with water and dried by nitrogen blow. Then, these cylindrical capillaries used for injecting and collecting fluids were coaxially assembled into a square capillary with an ID of 1.0 mm. Capillaries and tees were bonded and sealed using acrylate adhesive. The microfluidic device was mounted on a microscope stage. Each fluid was supplied through a silicone tube connecting the corresponding capillary to a syringe pump (LSPO2-1B, Longer, Hebei, China).

### 2.3. Characterization of Droplets and Microcapsules

Droplet generation in the capillary microfluidic device was visualized and recorded using a high-speed camera (Phantom MIRO M110, Vision Research, Wayne, NJ, USA) mounted on a microscope (CKX41, Olympus, Tokyo, Japan). Bright-field images were captured using a fluorescence microscope (Olympus IX2, Tokyo, Japan). Particle size was measured and analyzed using the software integrated with the microscope and the fluid flow rates. Field emission scanning electron microscopy (FE-SEM) (ZEISS Gemini 500, Carl Zeiss, Germany) was used to analyze the morphology and shell of microcapsules. 

### 2.4. Leakage Measurement of Microcapsules

The leakage of oxalic acid was measured using 913 acidometer (Metrohm, Switzerland) at room temperature (approximately 25 °C). Firstly, double-emulsion droplets within a continuous phase were collected from the outlet of the capillary device for 5 min. DI water was added into the dispersion to achieve a total volume of 10 mL. Then, the droplets in the dispersion were broken (demulsified) using a KQ-50DE ultrasonic cleaner (Ultrasonic Instruments Co., Ltd., Jiangsu, China). Afterwards, the demulsified dispersion was allowed to stand for 1 h at room temperature, ensuring that all core material (oxalic acid) was released into the continuous aqueous phase. pH value of the dispersion was measured using a pH acidometer. The total amount of released oxalic acid was calculated based to the first ionization equilibrium constant K_a1_ = 5.9 × 10^−2^. The second ionization equilibrium constant of oxalic acid (K_a2_ = 6.4 × 10^−5^) and pH of PVA solution (5.86 at room temperature) were both ignored. For each sample (data point), this process was repeated three times.

The same process was applied to the sample containing microcapsules instead of droplets. The dispersion containing droplets collected from the microfluidic device was exposed to UV light (INTELLI-RAY 400, Uvitron International, Inc., West Springfield, MA, USA) to cross-link the middle phase to form shells of the core–shell microcapsules. DI water was added into the dispersion to achieve a total volume of 10 mL before pH measurement. pH value was measured at periodical intervals to calculate the amount of leaked oxalic acid from microcapsules. The amount of leakage from microcapsules was compared to that from droplets (as total amount of encapsulation) to obtain the leakage percentage. For each sample (data point), such process was repeated three times.

### 2.5. Mechanical Strength Measurement of Microcapsules

In order to characterize the usability of microcapsules, the mechanical strength was measured. A total of 100 microcapsules were selected under the microscope and transferred to the surface of wood (loss error <5% during transfer). Afterwards, the other wood was put onto the surface, gradually applying pressure to the front side using a tension meter (HP-200, Edberg Instruments, Zhejiang, China). The pressure at which all microcapsules ruptured was considered as the mechanical strength of the microcapsules. In order to secure the reproducibility, each test was performed three times to obtain the averaged value. 

### 2.6. Thermogravimetric Analysis

Thermogravimetric/synchronous differential thermal analyzer (TGA/SDTA, Mettler Toledo, Switzerland) was used to perform thermogravimetric analysis (TGA) in nitrogen atmosphere at heating rate of 2 °C/min. 

### 2.7. Viscosity Measurement

In order to investigate the effect of added microcapsules containing oxalic acid on UF resin, a digital rotary viscometer (NDJ-5S, Shanghai Geology Instrument Institute, Shanghai, China) was used to measure the viscosity of pure UF resin adhesive, the mixture of UF resin and oxalic acid solution, and the mixture of UF resin and microcapsules containing oxalic acid, respectively. When mixing microcapsules or oxalic acid with UF resin, the amount of oxalic acid encapsulated in the microcapsules was kept equal to the amount of the added pure oxalic acid solution.

### 2.8. Bonding Strength Measurement

In order to verify the practical effect of the microcapsules containing curing agent in PSA, a bonding test was implemented using real wood boards. The adhesive formulation consisted of 10 parts UF resin powder, 4 parts water, and 0.5–2.5 parts microcapsule curing agent. Boards with a length of 100 mm and width of 25 mm were chosen, and a coated glue area of 25 × 25 mm^2^ was fixed for each time to ensure precise quantification. The PSA mixture was coated to wood board surface with a coating amount of 300 g/m^2^ and then covered by the other wood board. Then, the sandwiched PSA was pressed by applying a pressure of 2.0 MPa for 10 min on the top surface, and then stored for at least 12 h before measurement. All experiments were carried out under ambient conditions in a lab. 

## 3. Results

### 3.1. Double-Emulsion Droplets Formation in Microfluidic Devices

Figure 1a shows the schematic working mechanism of the capsule-based PSA. Microcapsules containing active reagents were mixed with UF resins and then coated on the surface of wood. When the wood was pressured, the microcapsules were ruptured to release core materials, which initiated a cross-linking reaction to achieve bonding. Conventional PSA commonly contained capsules with size in a wide range of a few to hundreds of microns or even wider, from nanometers to millimeters. As a result, at a fixed gap distance between two surfaces, only those capsules with an outer diameter larger than the gap distance could be ruptured to release the core materials (top row of Figure 1a). For practical application, capsules with well-controlled size and uniformity were highly required to achieve the simultaneous release and even distribution of core materials for higher bonding performance (bottom row of Figure 1a).

Using the microfluidic technology, monodisperse W_in_/O_mid_/W_out_ double-emulsion droplets could serve as templates for constructing microcapsules in a controllable manner, as shown in Figure 1b. To ensure the stable production of W_in_/O_mid_ droplets, the capillary for injecting the middle phase was hydrophobized [29]. A co-flow capillary microfluidic device was assembled to produce double-emulsion droplets in two emulsification stages. The droplet generation mechanism can be mainly categorized into dripping and jetting regimes [23]. At the 1^st^ stage, the inner phase of aqueous solution containing oxalic acid was emulsified to generate W_in_/O_mid_ emulsion droplets. Consequently, these droplets were further emulsified to produce (W_in_/O_mid_)/W_out_ double-emulsion droplets. At the micron scale, the capillary number, $Ca=\frac{\mu V}{\sigma }$ (where *µ* is the viscosity of continuous phase, *V* is the shear rate on dispersed phase, and *σ* is the interfacial tension between the two fluidic phases), is applied to describe multiphase fluidic flow [30]. Appendix A lists the measured viscosity and interfacial tensions of the inner, middle, and outer phases. *Ca* of 10^−4^–10^−2^ was typically obtained corresponding to the applied flow rates of 1–100 µL/min, representing laminar flow in the microfluidic device.

In the microfluidic device, the emulsion patterns and droplet size could be precisely controlled by adjusting the flow rates of inner, middle, and outer phases. Experimentally, in the range for producing stable double-emulsion droplets, we found four specific patterns, namely Type I, Type II, Type III, and Type IV, as shown in Figure 2a. In Types I and II, stable double-emulsion droplets (one middle phase droplet containing one inner droplet) were obtained with a shell thickness (*λ*) of 12–30 and <12 µm, respectively. At Types III and IV, the double-emulsion droplets could not encapsulate specifically just one inner phase droplet, but either ≥1 (1 or 2) or ≤1 (0 or 1) droplets in one encapsulation, respectively. Figure 2b,c present the diagram of flow patterns corresponding to the flow rates. When inner and middle phases flowed stably in the range of *Q*_in_ = 1–10 µL/min and *Q*_mid_ = 1–10 µL/min, the inner and middle phases could be encapsulated, respectively, at the 1st and 2nd stages. Thus, a Type I pattern was obtained by producing monodisperse double-emulsion droplets. At high *Q*_out_ and *Q*_in_, and low *Q*_mid_, larger inner phase droplets was surrounded by the middle phase at the 1st emulsification stage, and thus ultra-thin shell droplets (Type II) were formed at the 2nd emulsification stage. On the other hand, by decreasing the ratio of *Q*_in_ to *Q*_mid_, smaller and faster W_in_/O_mid_ droplet generation was obtained at the 1st stage; as a result, more than one inner droplet could be encapsulated at the 2nd stage. With the further increase of *Q*_out_, the droplet generation frequency at the 2nd stage was higher than that at the 1st stage, eventually resulting in the occurrence of the interlaced one or none encapsulation of inner phase droplets.

As discussed above, the microcapsule size, the size distribution, and the shell thickness are all important for PSA application; thus, the droplet inner diameter (*d*_in_), outer diameter (*d*_out_), and shell thickness (*λ*) corresponding to flow rates were investigated. Figure 3 presents *d*_in_, *d*_out_, and *λ* varying with *Q*_in_ (at fixed *Q*_mid_ and *Q*_out_), *Q*_mid_ (at fixed *Q*_in_ and *Q*_out_), and *Q*_out_ (at fixed *Q*_in_ and *Q*_mid_), respectively. Typically, the droplet diameter was in the range of 200–330 µm, and the obtained shell thickness was in the range of 5–35 µm. The overall *d**_in_* and *d**_out_* increased with *Q*_in_, and *λ* decreased with *Q*_in_, as demonstrated in Figure 3a, because of the occupation of inner core in the microcapsules. *λ* increased obviously with *Q*_mid_ at fixed *Q*_in_ and *Q*_out_, as shown in Figure 3b. At the same *Q*_mid_ and *Q*_in_, higher *Q*_out_ would squeeze the middle phase and produce a thinner shell; thus, *d*_in_, *d*_out_, and *λ* decreased, as demonstrated in Figure 3c. The size of the inner core depended on the 1^st^ emulsification stage, and the increase of the outer phase could only reduce the outer diameter of the double-emulsion droplets during the 2nd emulsification stage. Therefore, *d*_in_ did not change obviously with *Q*_out_ at constant *Q*_in_ and *Q*_mid_. However, when *Q*_out_ was increased to 40 µL/min, a sudden decrease in *d_in_* was induced from the change in droplet generation patterns (Figure 2b, green triangle).

### 3.2. Microcapsule Characterization

When the stable generation of double-emulsion droplets was achieved in a microfluidic device, the double-emulsion droplets were polymerized by UV into core–shell microcapsules. The monomer structure of ETPTA used for the capsule preparation is drawn in Appendix A. Figure 4a shows the monodisperse core–shell microcapsules with outer diameters of 275 ± 10 µm and the shell thickness of 20 ± 3 µm. Figure 4b–d show the hollow core–shell structure with a relatively smooth surface.

The microcapsules could encapsulate active components to achieve an extended lifetime for storage, controlled release to achieve on-demand operation, and pressurized mechanical rupture for ease of use. Effective factors for microcapsule-based PSA include the capsule size and size distribution, the core-material encapsulation efficiency, and the shell permeability and mechanical strength. Encapsulation efficiency and particle size distribution determine the utilization efficiency of microcapsules, and the permeability and mechanical properties of the shell directly affect the storage condition, shelf life, and usability of material. Typically, these microcapsules prepared using the microfluidic device could achieve >90% encapsulation efficiency with variable size in the range of 100–350 μm, and excellent monodispersity with the coefficient of variation (CV) of <5% [5,31,32,33].

To investigate the effect of shell thickness on oxalic acid leakage, we prepared microcapsules with similar sizes (*d*_out_) but different shell thicknesses (*λ*). pH values were measured at periodical intervals to obtain the leakage ratio of the microcapsules. As shown in Figure 5a, 0.2%–1.3% oxalic acid had been leaked into the dispersion before the pH measurement. This may be due to the demulsification of some droplets during the collection and curing processes. Afterwards, driven by osmotic pressure across the shell, the oxalic acid gradually penetrated through the shell to the outer phase. For comparison, the microcapsules with different *λ* were studied. As shown in Figure 5a, in 180 min, about 5.2%, 1.2%, and 4.0% oxalic acid was released from the microcapsules with *λ* of 14.6 ± 2, 20.5 ± 3, and 31.1 ± 5 µm, respectively. For the microcapsules with a thin shell of *λ* = 14.6 µm and thick shell of *λ =* 31.1 µm, the leakage rate for both was fast during the first 30 min. When the shell was too thin, the expansion or contraction from an external force or environmental change could induce shell leakage easily, thus causing a small amount of microcapsules to rupture (Figure 5b). On the other hand, when the shell was too thick compared to the core size, cracking was also more likely to occur (Figure 5c) according to the internal phase deviation from the spherical center [34]. As the kernel deviated from the center, the middle phase was distributed unsymmetrically; thus, the actual shell thickness on one side of the microcapsule was greatly reduced. When the shell material was solidified, the stress on the inner core was uneven, and the thinnest point of the shell tended to be more susceptible to breakage and leaking. As a result, the slowest and least leakage was obtained for the microcapsules with moderate shell thickness of *λ* = 20.5 µm (Figure 5d). 

A microcapsule-based PSA is highly convenient for practical usage since it just needs a pressure force to rupture the microcapsule shells. Therefore, the mechanical strength of shells should be evaluated to avoid soft shell induced unrupturable microcapsules. The mechanical strength of all prepared microcapsules with various shell thicknesses was higher than 0.57 N, as shown in Figure 6a. This value was seven times higher than the sodium alginate chitin shells from the self-healing capsules [35]. The mechanical strength increased with the microcapsule shell thickness, as expected. This also proves that such microcapsule-based PSA could withstand standard mechanical stirring in large industrial equipment. Considering the three aspects of the encapsulation volume, the leakage, and the mechanical strength, microcapsules with a shell thickness of about 20 ± 3 µm were chosen for testing and practical use.

Thermogravimetric analysis (TGA) was applied to characterize the thermal stability of the microcapsules. As shown in Figure 6b, three significant mass loss ranges were observed at 50–110, 120–250 and 360–470 °C, corresponding to the evaporation of water, evaporation (or decomposition) of oxalic acid glycol solution, and decomposition of the shell material, respectively. These three major mass loss distributions were compared by the measured curves of corresponding oxalic acid glycol solution, pure ethylene glycol, and microcapsule shell material, respectively. Experimental measurements show that there were about 3 wt% water, 41–42 wt% oxalic acid solution, and 52–55 wt% shell material in the microcapsules. These results are consistent with the calculated portion of oxalic acid solution in the microcapsules of 44 wt%, according to *Q*_in_ of 4 μL/min and *Q*_mid_ of 5 μL/min for producing the microcapsules in microfluidic device.

In order to simulate practical situations, we have also tested the viscosity change of UF resin (10 parts UF resin power mixed with 4 parts water) with different curing agents. The UF resin reaction mechanism is shown in Appendix A. As shown in Figure 6c,d, the initial viscosity of the UF resin was about 5.2 Pa·s, which increased slightly within 30 min. With the addition of oxalic acid, its viscosity rapidly increased with time (blue line) according to the polymerization reaction. The viscosity of the mixture of UF resin and microcapsules (with oxalic acid cores) was only slightly higher than that of the pure UF resin. This means that the slight oxalic acid leakage from the microcapsules did not affect the bonding performance. As a result, we can conclude that wrapping the oxalic acid by the microcapsules could prolong the pot life of the PSA.

### 3.3. Bonding Performance of the Microcapsule-Based PSA

During the bonding process, a pressure of 2.0 MPa was applied onto a plywood board for pressurizing the microcapsules sandwiched between two boards. With the addition of microcapsules up to 15 wt% in the PSA mixture, the applied pressure acting on the microcapsules could completely rupture the microcapsules to release the encapsulated curing agent (oxalic acid). The surface roughness *R*_z_ of the plywood surfaces was below 100 µm (Appendix A), which is consistent with the literature result [36]; therefore, the microcapsules with *d*_out_ of 200–300 µm were appropriate for the wood surface bonding operation. Moreover, the microcapsules prepared via microfluidic technology were highly monodisperse with CV <5%. All these features ensure that the applied pressure could rupture most of the microcapsules between two boards.

The bonding strength of the PSA based on microcapsules was evaluated by separating the bonded plywood boards. Appendix A shows the schematic of the bonding strength measurement. Figure 7a,b present the bonding strength varying with the amount of added microcapsules and the retention time. When the weight concentration of microcapsules in the PSA mixture was increased from 3.45 to 9.68 wt%, the bonding strength gradually increased from 0.73 to 0.91 MPa. However, with the further addition of microcapsules, the average bonding strength decreased to 0.66 MPa when the microcapsule concentration increased to 15.15 wt%. This might be explained by the increased portion of the remaining shell materials in the PSA. As a result, the effective bonding area (portion) of the PSA layer was reduced, and the bonding strength was decreased. 

In order to verify the long-term effectiveness of the microcapsules in practical conditions, we have also placed the mixture of microcapsules and UF resin for a period of time before coating. The bonding strength slightly increased with the retention time up to 3 h, as shown in Figure 7b. This may be attributed to the viscosity increase induced by water evaporation. On the other hand, it also suggests that the trace amount of oxalic acid leakage before rupture did not affect the bonding efficiency of PSA. The experimentally measured bonding strength was all above 0.7 MPa, satisfying the standard for plywood bonding processing (GB/T 9846-2015). Figure 7c,d show SEM images of the ruptured microcapsules in the PSA after polymerization. Ruptured microcapsules were wrapped in the PSA to achieve bonding. In summary, the prepared core–shell microcapsules have been successfully validated for practical application in PSA for high-performance wood boards bonding, showing advantages of easy usage, efficient material consumption, and high bonding strength.

## 4. Conclusions

In conclusion, we have proposed and validated the concept of microfluidic construction of high-quality microcapsules for pressure-sensitive adhesive with enhanced performance. The core–shell microcapsules prepared from double-emulsion droplet templates using a microfluidic device showed high monodispersity with controllable diameter and shell thickness. The microcapsules showed high stability with low leakage rate over time in water and a steady mechanical strength of >0.57 N. Mechanically ruptured microcapsules released the core material of oxalic acid to initiate polymerization with mixed UF to achieve bonding performance. The typical bonding strength over 0.7 MPa has been achieved by the PSA from mixing microcapsules with water and UF, proving its practical applicability in industry. The encapsulation of reactive components cannot only prolong the pot life of PSA materials, but it can also achieve controlled release as required for practical applications. The size uniformity ensures high rupture efficiency of the microcapsules and an even distribution of core materials during compressing. With further tuning in capsule size and material compositions, these microcapsules-based PSA could be applied in finer processes of modern electronic, optical, and biomedical devices. 

## Figures and Tables

**Figure 1 nanomaterials-10-00274-f001:**
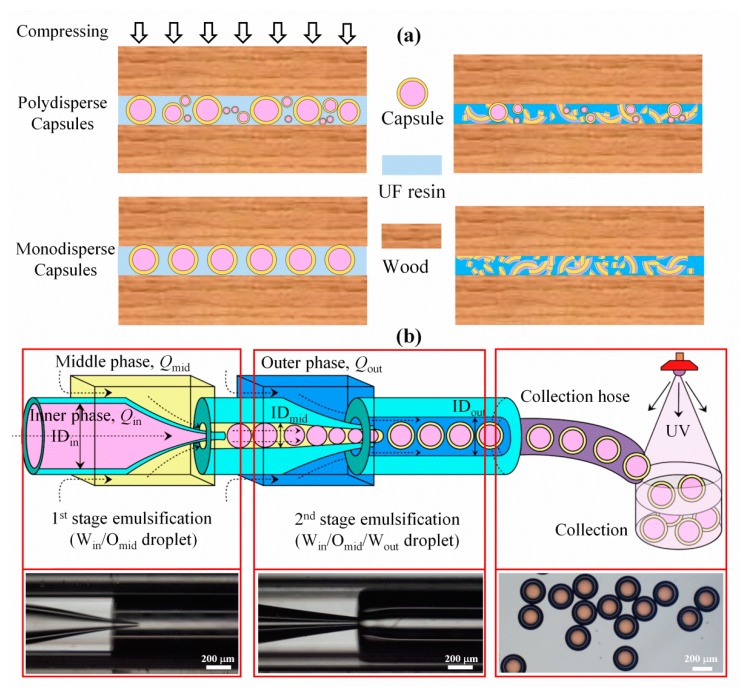
(**a**) Schematic working mechanism of the capsules-based pressure-sensitive adhesive (PSA) with polydisperse (top) and monodisperse (bottom) microcapsules. (**b**) Schematic (top row) and optical images (bottom row) showing the step-by-step process of constructing monodisperse microcapsules using a capillary microfluidic device for preparing the double-emulsion droplets, and UV irradiation for polymerizing the middle phase to form the shells to obtain microcapsules.

**Figure 2 nanomaterials-10-00274-f002:**
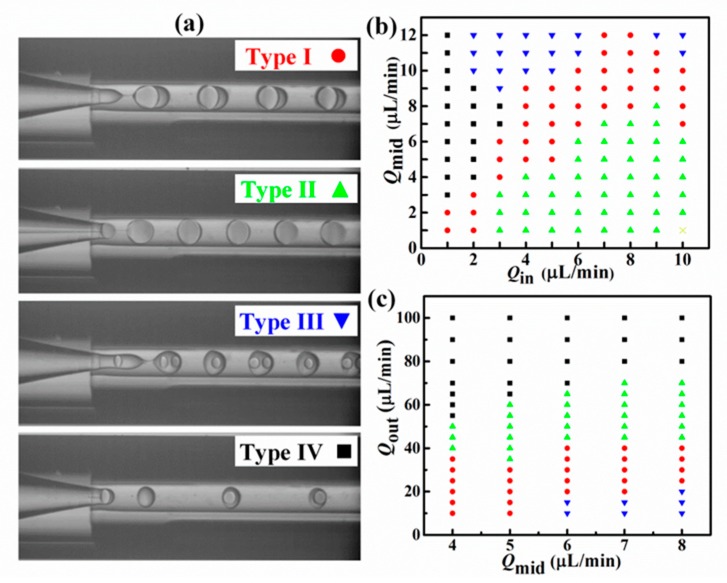
Double-emulsion droplets formation in a capillary microfluidic device. (**a**) Four types of droplet formation patterns marked with Type I (red circle), Type II (green triangle), Type III (blue inverted triangle), and Type IV (black square), denoting normal monodisperse double-emulsion droplets, double-emulsion droplets with an ultra-thin shell, more than one type of inner droplet encapsulated in one droplet, and one or none inner droplets encapsulated in one droplet, respectively. (**b**) Diagram of droplet patterns varying with *Q*_in_ and *Q*_mid_ at constant *Q*_out_ = 30 µL/min. (**c**) Diagram of droplet patterns varying with *Q*_mid_ and *Q*_out_ flow rates at (*Q*_mid_ − *Q*_in_) = 1 µL/min.

**Figure 3 nanomaterials-10-00274-f003:**
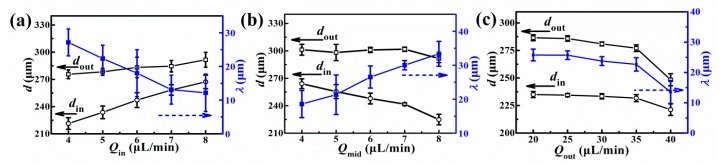
Droplet diameter (*d*_in_ and *d*_out_) and *λ* varying with (**a**) *Q*_in_, at *Q*_mid_ = 5 µL/min and *Q*_out_ = 30 µL/min, (**b**) *Q*_mid_, at *Q*_in_ = 5 µL/min and *Q*_out_ = 30 µL/min, and (**c**) *Q*_out_, at *Q*_in_ = 5 µL/min and *Q*_mid_ = 6 µL/min.

**Figure 4 nanomaterials-10-00274-f004:**
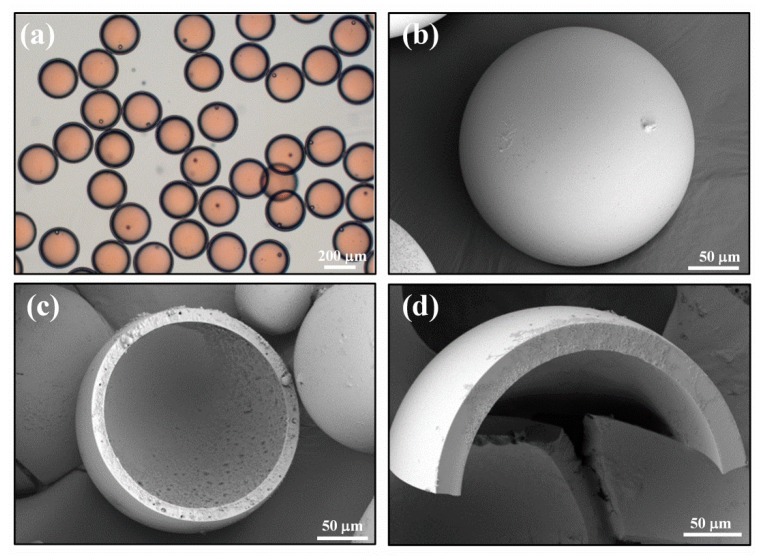
(**a**) Optical microscopy of monodisperse core–shell microcapsules after curing. SEM images of (**b**) an integral microcapsule, (**c**) a ruptured microcapsule, and (**d**) close-up view of a microcapsule’s shell.

**Figure 5 nanomaterials-10-00274-f005:**
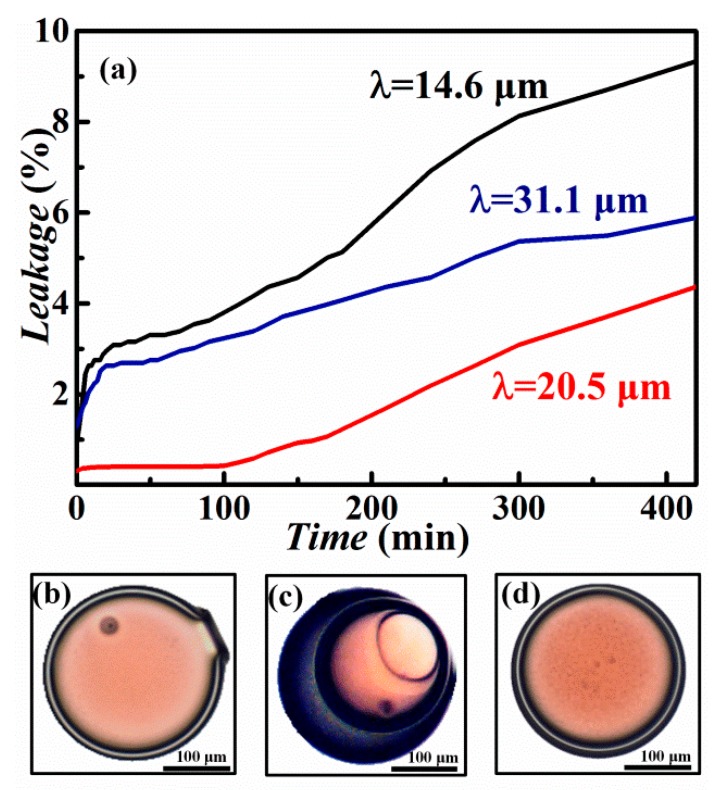
(**a**) Release profile of oxalic acid from microcapsules with different shell thicknesses of 14.6, 20.5, and 31.1 µm. (**b**) Microscopy image of a prematurely ruptured microcapsule with *λ* = 14.6 µm (thin shell). (**c**) Microscopy image of a microcapsule with *λ* = 31.1 µm (asymmetric shell thickness). (**d**) Microscopy image of a microcapsule with *λ* = 20.5 µm (uniformly distributed shell thickness).

**Figure 6 nanomaterials-10-00274-f006:**
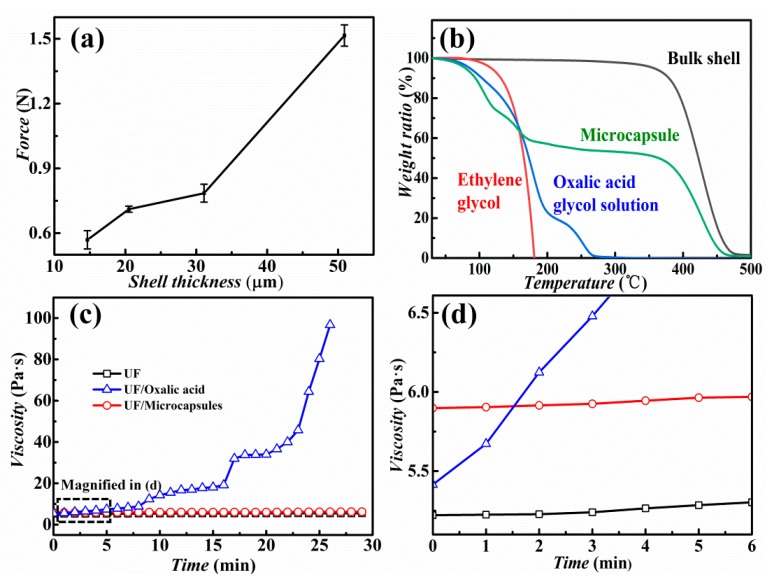
(**a**) Mechanical strength of microcapsules varying with shell thickness. (**b**) Thermogravimetric analysis (TGA) curves of the microcapsules, the bulk shell material, ethylene glycol, and the oxalic acid glycol solution. (**c**) Viscosity change with time for the pure urea–formaldehyde (UF) resin, the mixture of UF resin and oxalic acid, and the mixture of UF resin with microcapsules with oxalic acid cores. (**d**) Enlarged area for close-up view in the dotted rectangle in (**c**).

**Figure 7 nanomaterials-10-00274-f007:**
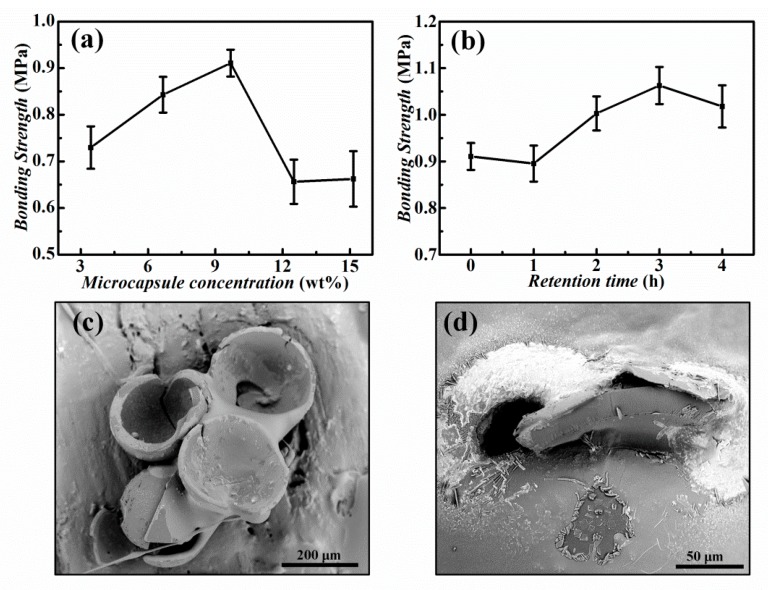
(**a**) Bonding strength of the wood boards versus the concentration of microcapsules in pressure-sensitive adhesive (PSA) mixture. (**b**) Bonding strength of the wood boards versus the retention time of the PSA containing 9.68 wt% microcapsules. (**c**) SEM image of ruptured microcapsules in PSA after polymerization. (**d**) SEM image of a ruptured capsule wrapped in the PSA.

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
