# Peer review of "Microfluidic-Assisted Fabrication of Monodisperse Core–Shell Microcapsules for Pressure-Sensitive Adhesive with Enhanced Performance"

_nanomaterials, 2020, doi:10.3390/nano10020274_

Round 1

Reviewer 1 Report

The paper provides very interesting results; however, the authors should:

Line 40, the sentence: Encapsulation of materials is normally for isolation or protection, being evolved from nature, ranging from nanoscale to macroscale, from a cell to an egg. Is missing a verb (i.e. is normal used, applied, etc.)

It looks like different parts of the manuscript are written by different contributors, sometimes the text is in Present Simple or past simple, please uniform it.

The authors should provide the monomer structures used for capsules preparation.

Line 235: the authors should add an error range: Typically, the droplet diameter was in the range of 200-330 μm, and the obtained shell thickness was in the range of 5-35 μm.

Same in line 254, 276, 301, etc…. all results should be provided with an error range.

Line 266 references are missing in the sentence: Typically, these microcapsules prepared using the microfluidic device could achieve >90% encapsulation efficiency with variable size in the range of 100-350 m, and excellent monodispersity with the coefficient of variation (CV) of <5%. (at least 3 references should be added)

Reviewer 2 Report

The paper entitled “Microfluidic-Assisted Fabrication of Monodisperse Core-Shell Microcapsules for Pressure-Sensitive Adhesive with Enhanced Performance” provides an interesting study on the microfluidic construction of core-shell microcapsules containing oxalic acid for the realization of pressure-sensitive adhesives. The microcapsules were prepared through the UV-polymerization of double-emulsion droplet templates and exhibited high monodispersity and good encapsulation properties. Moreover, a deep investigation on the bonding performance of pressure-sensitive adhesives containing the core-shell microcapsules was carried out, by mixing microcapsules with water and UF powder and depositing the mixture between two wood boards, thus proving the high potential of the designed system also at industrial scale. The general concepts of the work are straightforward and clever, and the results are of interest for the people working in the field of microencapsulation and self-healing materials. Moreover, the English is adequate, and references are appropriate. Therefore, I think the paper is worth to be published in Nanomaterials as it is.

Author Response

We sincerely appreciate the reviewer for your evaluation and recommendation for publication of our work in Nanomaterials.

Round 2

Reviewer 1 Report

The improved manuscript can be accepted.